# PALS: Personalized Active Learning for Subjective Tasks in NLP

**Kamil Kanclerz**[1], **Konrad Karanowski**[2], **Julita Bielaniewicz**[1], **Marcin Gruza**[1],
**Piotr Miłkowski**[1], **Jan Kocoń**[1], **Przemysław Kazienko**[1]
Wrocław University of Science and Technology
Wrocław, Poland
[1]{kamil.kanclerz,julita.bielaniewicz,marcin.gruza,
piotr.milkowski,jan.kocon,kazienko}@pwr.edu.pl
[2]254533@student.pwr.edu.pl

## Abstract

For subjective NLP problems, such as classification of hate speech, aggression, or emotions, personalized solutions can be exploited. Then, the learned models infer about the perception of the content independently for each reader. To acquire training data, texts are commonly randomly assigned to users for annotation, which is expensive and highly inefficient. Therefore, for the first time, we suggest applying an active learning paradigm in a personalized context to better learn individual preferences. It aims to alleviate the labeling effort by selecting more relevant training samples. In this paper, we present novel Personalized Active Learning techniques for Subjective NLP tasks (PALS) to either reduce the cost of the annotation process or to boost the learning effect. Our five new measures allow us to determine the relevance of a text in the context of learning users' personal preferences. We validated them on three datasets: Wiki discussion texts individually labeled with aggression and toxicity, and on the Unhealthy Conversations dataset. Our PALS techniques outperform random selection even by more than 30%. They can also be used to reduce the number of necessary annotations while maintaining a given quality level. Personalized annotation assignments based on our controversy measure decrease the amount of data needed to just 25%-40% of the initial size.

## 1 Introduction

The data acquisition process plays a crucial role in almost all natural language processing (NLP) studies. The relevant data can significantly improve the model performance. However, the low-quality data limits the model quality regardless of its advanced architecture. Modern machine learning systems require large amounts of data to perform satisfactorily. Although deep learning rapidly advances state-of-the-art results on a number of supervised learning tasks (Mukherjee and Awadallah, 2020; Kocoń and Maziarz, 2021; Korczyński and Kocoń, 2022; Kocoń et al., 2022; Srivastava et al., 2023; Koptyra et al., 2023), we can observe that gains strongly depend on large annotated datasets (Kanclerz et al., 2020; Shim et al., 2021; Kocoń et al., 2021b,a). Moreover, the data annotation process is very expensive (Kocoń et al., 2019a,b; Wierzba et al., 2021). Several lines of research, especially active learning, have developed mechanisms to reduce the amount of supervision required to achieve better predictive performance. They, however, focused on tasks themselves, i.e., such text selection that would increase agreement among annotators. This does not seem to be adequate for subjective NLP tasks, in which we allow the model to provide different decisions for different users and the same text (Kocoń et al., 2023a; Mieleszczenko-Kowszewicz et al., 2023). As a result, there are no active learning methods dedicated for subjective tasks and personalized reasoning.

Therefore, we introduce a new approach to active learning in NLP: PALS - Personalized Active Learning for Subjective problems. Its main contribution is modeling user preferences (perception) in the context of a given task. This allows us to exploit existing personalized models (Kocon et al., 2021; Kocoń et al., 2023b; Kazienko et al., 2023) and complement them with more appropriate data acquisition. The main idea of PALS is the selection of texts for annotation adapted to a given user that maximizes extraction of knowledge about the user preferences. It has been achieved by means of five measures described in Sec. 4. Our measures not only improve the overall performance of personalized models. They also significantly decrease the number of annotations needed. As a result, they reduce computing power, which saves money and cuts down the environmental toll. The final product of our PALS methods is the optimization of the data collection process, which allows us to reduce the cost. Less but more adequate data may provide the same training effect. Alternatively, we

acquire more information for the same cost. Additionally, the annotation process can be stopped after achieving desirable quality. To develop efficient PALS methods, we raised several questions. How many annotations should each participant evaluate? Should we focus more on controversial or unambiguous texts?

Please note that currently there is no method for personalized selection of texts for annotation appropriate for subjective tasks – PALS is the first.

Our main contribution in this work are: (1) posing a new problem regarding the effective acquisition of annotations from users in order to acquire knowledge about both the task and individual user perception supporting the learning of personalized reasoning models for subjective NLP tasks, (2) our novel PALS solution – five measure-based methods, which capture the diversity in subjective text perception, (3) validation of our PALS methods on three datasets and ten classification tasks related to the offensiveness of texts for different number of documents annotated by the user, (4) ablation study including the analysis of what fraction of the training set is sufficient to achieve performance comparable to a reference random selection.

## 2 Related work

Active Learning (AL) has been extensively studied and applied in numerous tasks over recent years. The costs and means of obtaining labeled datasets can be troublesome, and this fact became more prominent as the research in the domain of machine learning began to develop rapidly. The need of training classifiers using the least amount of data sprouted nearly 20 years ago, yet grew stronger in recent years. The more common AL approaches use sampling based on model uncertainty using measures such as margin sampling (Zhou and Sun, 2014; Prabhu et al., 2019), least model confidence (Lowell et al., 2018) and entropy (Zhu et al., 2008; Zhang et al., 2017). Deep Learning in AL, also named Deep Active Learning, can be a challenging task due to the tendency of rare uncertainty during inference and the requirement of a relatively large amount of training data. However, Bayesian Deep Learning based approaches (Wang et al., 2016; Tan et al., 2021; Ren et al., 2021) proved that using dropout, batch normalization, and ensembles were all useful ways of lowering uncertainty in model prediction. There have been numerous promising implementations of AL in the field of Natu-

ral Language Processing. A significant amount of literature showed that the previously mentioned Bayesian Active Learning outperformed classical uncertainty sampling. However, the newest approaches (Lu et al., 2019; Prabhu et al., 2021) explored AL with multiple strategies using BERT based models for binary text classification tasks and proved that the performance of BERT-based models managed to achieve even better scores. Visual Active Learning (Harpale and Yang, 2008; Wang et al., 2016) is a generally known domain of AL, which is explored in learning-based image classification. Common implementations incorporate deep convolutional neural networks into AL. The sample selection strategy works as a cost-effective improvement of the classifier through incorporating the analysis of high-confidence samples from an unlabeled set for feature learning. The selection of high-confidence samples leads to an automatic selection and interactively assigned pseudo labels.

The vast majority of AL work bases its mechanisms and score comparison on the generalized approach. The first attempts at developing personalization in AL extended the Bayesian AL techniques (Harpale and Yang, 2008; Seifert and Granitzer, 2010) through computing the probability of getting any rating, instead of the probability of getting a particular rating for a text. The algorithm is aimed at selecting items which have a very high probability of getting a rating from a certain user. This approach provides interesting scores, however, keeping the gold standard as a generalized measure. Therefore, the area of personalization remains dependent on generalization to a certain degree (Bernard et al., 2018; Kocoń et al., 2023a; Mieleszczenko-Kowszewicz et al., 2023; Ferdinan and Kocoń, 2023). What we achieved in our work is the contrary, it is the person themselves who becomes the gold standard, through analyzing the annotations and ratings they submitted. This breakthrough not only may change if we analyze users' annotations in the AL domain, but also in analyzing data universally.

## 3 Generalized vs. Personalized Active Learning for Text Annotation

Active learning techniques should take into account the nature of the task to which they are applied. In the case of personalized classification, this task requires the model to combine the general knowledge about the task (nature of the text) with the

individual preferences of the user (individual content perception). Here, this knowledge balance is crucial, as a model too focused on the user will not take into account the linguistic aspects of the text. On the other hand, a classifier mainly exploiting linguistic knowledge will not be able to make personalized predictions given the user's point of view. To ensure this balance, we suggest the introduction of personalized active learning techniques. They should take into account the impression that the text evokes in the user and how much knowledge about the user's preferences should be gathered. To reveal individual user beliefs, a text selected for annotation must be controversial and the entire set of texts assigned to the user should contain contrasting examples.

Text subjectivity can be measured both in generalized and personalized ways. In the first case, we examine how diverse are the opinions that the text evokes among all annotators. The generalized ambiguity of a text can provide information about its potential to gain knowledge about the personal preferences of each annotator. However, this approach can mask detailed information about the preferences of individual people with unique points of view by treating all annotators holistically. To counteract this, knowledge of the general text controversy should be enriched with personalized information about the user's past decisions to assess how their individual subjectivity compares with the generalized subjectivity of other users (Kanclerz et al., 2021, 2022; Bielaniewicz et al., 2022; Miłkowski et al., 2022; Ngo et al., 2022).

Generalized Active Learning (GAL) methods for text annotation focus on selection of documents that better reflect a given NLP problem averaged over the whole user community without considering any individual user preferences. As a result, every user may annotate the same texts, Fig. 1. Usually their goal is to maximize the inter-annotator agreement level (Lin et al., 2019). For that reason, they do not appear to be suitable for subjective tasks like personalized offensiveness detection.

We, however, introduce another idea – Personalized Active Learning for Subjective problems (PALS), in which texts are selected for annotation individually for a particular user to better capture their specific content and task perception. In other words, we implement the human-centred perspective introduced in (Kocoń et al., 2021), which matches single users with subjective NLP tasks.

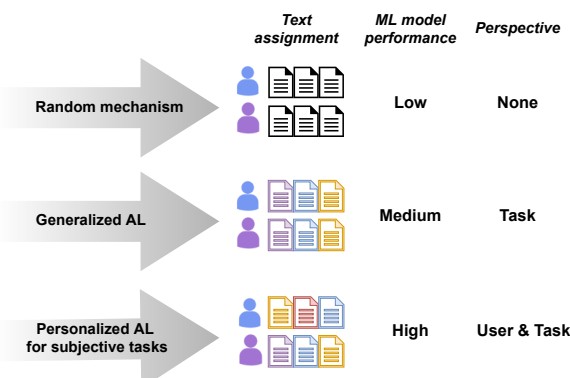

Figure 1: Random vs. Generalized Active Learning (GAL) vs. Personalized Active Learning for Subjective tasks (PALS) used to select texts to be annotated. The random approach (top) does not use any active learning or consider any perspective. GAL (middle) is oriented towards task itself (documents) providing the same texts to all users. In PALS - our personalized strategy (bottom), text selection also respects the individual user perspective.

This can further improve the performance of personalized architectures by reducing the number of redundant documents. Such user-specific knowledge also facilitates distinguishing one user from another in a more efficient way.

Since we focus on subjective tasks, we experimentally confronted our methods with the baseline (random selection) rather than with generalized approaches, which were not designed to infer for subjective problems.

## 4 Active Learning Mechanisms

We assume that, in the case of subjective tasks, active learning methods should respect user preferences. To achieve it, we prepared five measures: (1) Controversy, (2) Var Ratio, (3) Ratio Distance, (4) Stranger Count, and (5) Average Minimal Conformity. They take into account the diversity of the aggregated user beliefs or focus solely on the decisions of a single individual. Texts with the greatest value of a given measure are selected to be annotated by the particular user.

(1) **Controversy** ($Contr(d)$) of document $d$ is an entropy-based measure, as follows:

$$Contr(d) = \begin{cases} 0, \text{ if } n_d^0 = n_d \vee n_d^1 = n_d \\ -\sum_{c=0,1} \frac{n_d^c}{n_d} \log_2\left(\frac{n_d^c}{n_d}\right), \text{otherwise} \end{cases}$$

where $n_d^0$ and $n_d^1$ is the number of negative and positive annotations assigned to document $d$, respectively; $n_d$ – the number of all annotations related to

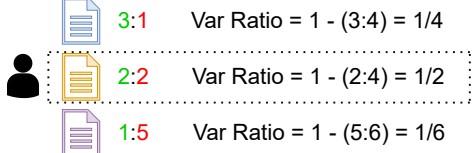

| | 3:1 | Var Ratio = 1 - (3:4) = 1/4 |
| | 2:2 | Var Ratio = 1 - (2:4) = 1/2 |
| | 1:5 | Var Ratio = 1 - (5:6) = 1/6 |

Figure 2: An example of Var Ratio metric calculation. The middle document has the highest score, meaning that it is the best candidate for annotation.

$d$, $n_d = n_d^0 + n_d^1$; $\frac{n_d^c}{n_d}$ estimates the probability that annotation of document $d$ assigns it to the class $c$. $Contr(d) = 0$ happens when all users assigned $d$ to the same class $c$, $Contr(d) = 1$ means that 50% of users considered it aggressive and 50% did not.

(2) The **Var Ratio** ($VarRatio(d)$) measure aims to calculate the dispersion around the class, to which the document ($d$) was assigned most often (mode). The main intuition is that the text is ambiguous if the majority class was assigned to the document relatively rarely. This often indicates that users annotated the document as belonging to the other classes a relatively large number of times. The Var Ratio measure is defined as follows:

$$VarRatio(d) = 1 - \frac{n_d^{mode}}{n_d}$$

where $n_d^{mode}$ is the number of users who annotated $d$ to its largest class. An example of Var Ratio usage is shown in Fig. 2.

(3) The **Ratio Distance** ($RatioDist(d,u)$) measure focuses on the difference between the ratio of users who annotated document $d$ as aggressive/toxic ($ratio(d)$) and the average ratio of users ($avg\_ratio(u)$), who assigned the document to the positive (aggressive or toxic) class ($c = 1$) for documents already annotated by the user $u$. In the first step, $ratio(d)$ is calculated for each document $d$ and annotations of users who annotated $d$ ($u \in U_d$). The number of annotations assigning $d$ to the positive class ($\mathbb{1}_{\{l_{d,u}=1\}}$) is divided by the total number of $d$'s annotations ($\mathbb{1}_{\{l_d \in C\}}$):

$$ratio(d) = \frac{\sum_{u \in U_d} \mathbb{1}_{\{l_{d,u}=1\}}}{\sum_{u \in U_d} \mathbb{1}_{\{l_d \in C\}}}$$

$$avg\_ratio(u) = \frac{\sum_{d \in D_u} ratio(d)}{|D_u|}$$

$$RatioDist(d,u) = ratio(d) - avg\_ratio(u)$$

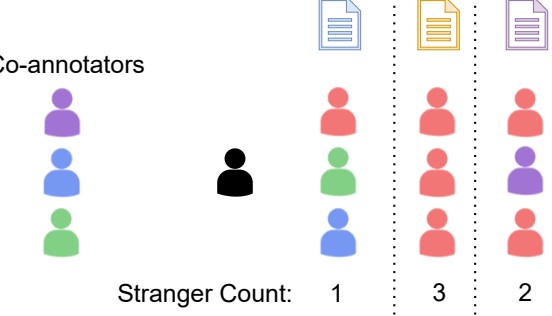

Figure 3: An example of Stranger Count measure calculation. Co-annotators are people who already annotated documents with our considered black user. On the right side, we have new red users, who have not yet annotated any document with our user. The greatest Stranger Count value identifies the orange text as the best to be annotated by our black user.

(4) The **Stranger Count** measure ($StrangerCount(d,u)$) aims to compare the subjective preferences of the user with as large groups of unique users as possible, Fig. 3. Its value for a specific document $d$ in the context of user $u$ is equal to the difference between the number of users who annotated $d$ ($|U_d|$) and the number of annotators ($a \in U_d$) whose set of annotated documents ($U_a$) contains at least one common document with the set of documents annotated by $u$: $|D_u \cup D_a| > 0$:

$$StrangerCount(d,u) = |U_d| - |\sum_{a \in U_d} \mathbb{1}_{\{|D_u \cap D_a|>0\}}|$$

(5) The **Average Minimal Conformity** ($AvgMinConf(d)$) describes the frequency of $u$ belonging to the majority of users over all $u$'s texts. Firstly, the conformity measure ($Conf(u,c)$) is calculated for user $u$ and a given class $c \in C$:

$$Conf(u,c) = \frac{\sum_{d \in D_u} \mathbb{1}_{\{l_d \in c \wedge l_d = l_{d,u}\}}}{\sum_{d \in D_u} \mathbb{1}_{\{l_d \in c\}}}$$

where $D_u$ – the set of texts annotated by $u$; $l_{d,u}$ – the label assigned by $u$ to $d$; $l_d$ – the $d$'s label obtained via majority voting. The final value of the Average Minimal Conformity measure for document $d$ averages the minimal conformity of user $u$ over all considered classes ($c \in C$) and for all users $U_d$ who annotated $d$:

$$AvgMinConf(d) = \underset{u \in U_d}{avg}\left(\underset{c \in C}{min}(Conf(u,c))\right)$$

## 5 Subjective Tasks

In our experiments, We used datasets obtained within the Wikipedia Detox project [1] and the Unhealthy Conversations dataset (Price et al., 2020). Table 1 contains their profile. Each dataset has been divided into three subsets: training, validation, and testing in the 60-20-20 proportion. However, the original split of the data means that our model is trained and tested on the same users but on different texts. In the real world, we are faced with the problem of evaluating the same texts but with respect to new users. For this reason, we had decided to split the dataset with respect to users as suggested in (Kocon et al., 2021; Milkowski et al., 2021), Fig. 4, Sec. 6.1.

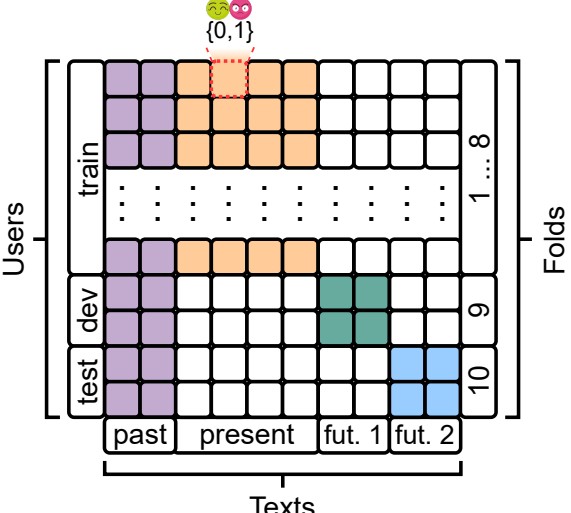

Figure 4: The U-T dataset split. Only the purple parts of the dataset are pruned in order to limit the amount of knowledge about the users in the dev and test folds the model can obtain

### 5.1 WikiDetox: Aggression and Toxicity

To ensure the best possible quality of Wikipedia entries, their authors discuss about them on the internal forum. Unfortunately, the forum has also become a place where some users verbally attack others. To counteract these phenomena, Wikimedia Research has developed methods to automatically detect aggressive, toxic, and offensive content (Wulczyn et al., 2017). The WikiDetox datasets were created on the basis of user comments on more than 95M articles in 2001-2005. As a result, three datasets were created. Since the *Attack* and

*Aggression* datasets are very similar in terms of user and annotation distribution, we selected two out of the three for our research: *WikiDetox Aggresison* and *WikiDetox Toxicity*. The labels of the original dataset are degrees of aggression/toxicity in the range [-3, 3]. Like other researchers, we decided to transform this into a binary classification task assuming that scores below 0 mean that the text is aggressive/toxic (class=1), and scores higher than or equal to 0 mean that the text is neutral (class=0).

### 5.2 Unhealthy Conversations

The Unhealthy Conversations dataset (Price et al., 2020) consists of 44k comments, the length of which does not exceed 250 characters. They were acquired from the Globe and Mail opinion articles obtained from the Simon Fraser University Opinion and Comments Corpus (Kolhatkar et al., 2020). Each comment is rated by at least three annotators, whose task was to assign at least one of the offensiveness labels: *antagonize*, *condescending*, *dismissive*, *generalization*, *unfair generalization*, *healthy*, *hostile* and *sarcastic*. Each comment was presented to annotators separately, devoid of context related to the article and other comments in order to reduce the bias.

## 6 Experimental Setup

In order to validate our personalized active learning methods, we have decided to apply the following principles. The experiments that were performed on all three datasets using the binarized versions of the tasks. Class imbalance was addressed by focusing on the positive, minority class {1} F1-score (F1-binary) metric.

### 6.1 Data preprocessing

In the data preprocessing, we removed users who annotated less than 40 texts and those who annotated all texts with the same label. This reduced the number of unique annotators to 3,719 for *WikiDetox Aggression* dataset and 4,128 for *WikiDetox Toxicity*. Additionally, we have removed all new line characters from the text and then, we tokenized them using XLM-RoBERTa dedicated tokenizer (Conneau et al., 2020).

We utilize the same user-text (U-T) data partition setup as in (Kocon et al., 2021; Milkowski et al., 2021).

The users were first divided into 10 folds. Similarly, the texts were split into 4 sets: *past*, *present*,

---

[1]https://meta.wikimedia.org/wiki/Research:Detox/Data_Release

| Output | WikiDetox: Aggression | WikiDetox: Toxicity | Unhealthy Conversations |
|---|---|---|---|
| Number of texts | 115,864 | 159,686 | 44,355 |
| Number of users | 4,053 | 4,301 | 558 |
| Number of annotations | 1,365,217 | 1,598,289 | 227,975 |
| Avg. annotations per user | 336.84 | 371.61 | 387.71 |
| Avg. annotations per text | 11.78 | 10.01 | 4.66 |
| Labels | {0, 1} | {0, 1} | 8 x {0, 1} |
| Language | English | English | English |

Table 1: Dataset profiles. Class=1 means the text is aggressive, toxic or unhealthy.

*future1*, and *future2*. Such partitioning allows us to test methods on texts and users that are unknown by the model. The *past* set represents the general knowledge about all users. This is the only part of the texts that are used to train the model with annotations made by users from test and validation folds. The *present* set is the main source of knowledge about the texts, while *future1* and *future2* represent the validation and test sets of texts.

## 6.2 Language Model

To obtain vector representations of texts, we used the XLM-RoBERTa model. It is a multilingual version of RoBERTa transformer trained on 100 languages and 2.5 TB of preprocessed Common-Crawl data.

Due to limited resources and the wide scope of experiments, the language model was not fine-tuned for downstream tasks during the experiments. Instead, we only used it as a feature extractor. The vector representation of the text is an average of the token vectors of the last transformer's layer. The only manual preprocessing step for the texts in the datasets was to remove new line token strings for Wiki datasets.

## 6.3 Classification models

As personalized classification models, we used HuBi-Simple, presented in (Kocon et al., 2021). HuBi-Simple is a relatively simple linear model that uses textual features and additionally learns the biases of annotators and biases of single words. The annotator bias is a straightforward component that allows us to capture the annotator personal bias towards the modeled NLP task. We selected this model because we want a personalized model to help us to evaluate our PALS approach to data annotation. We applied the same hyperparameter settings as proposed in the original model publication because our research covers two the same datasets and similar setup.

## 6.4 Experimental scenario

At the beginning, we took the entire *past* data from the training part. Then, at each step, we added to it one annotation from the set of *present* for each user, selected on the basis of our personalized active learning (PALS) methods, see Sec. 4. The procedure regarding the ordering of texts as candidates for annotation is described in Alg. 1. We trained our model on the set obtained in this way. Then, it was evaluated on the *future2* subset. We repeated this procedure until we reached the maximum available number of 14 annotations from the *present* set for each user. Each of such experiments was repeated 10 times in the U-T cross-validation scenario, Fig. 4. To examine the impact of data acquired with our PALS measures on the model performance, we confronted its results with the same model but trained on randomly annotated data (baseline).

---

**Algorithm 1** The process of ordering the texts – candidates for annotation $a_j \in A_u$ for user $u_i \in U$. The PALS measure $\psi$ value $m_j$ is calculated for each potential annotation $a_j$ not yet annotated by user $u_i$. Then, texts $a_j$ are sorted for a given user $u_i$ by the measure values.

---
1: $orders \leftarrow [[]]$
2: **for** $u_i \in U$ **do**
3:     $text\_measures \leftarrow \{\}$
4:     **for** $a_j \in A_u$ **do**
5:         $m_j \leftarrow \psi(a_j)$
6:         $text\_measures[a_j] \leftarrow m_j$
7:     **end for**
8:     $K = text\_measures.sort\_by\_values()$
9:     $orders[i] \leftarrow K.keys()$
10: **end for**
11: **return** $orders$

---

# 7 Results

In the case of *WikiDetox Aggression* (the upper plot in Fig. 5), the best results were achieved by the Var Ratio and Controversy selection. Average Minimal Conformity and Stranger Count demonstrated stable behavior, slightly different from the random annotations. However, the Average Minimal Conformity was clearly better in the range of 3-5 annotations per user. The Conformity Distance measure achieved significantly worse results than all other measures. Our experiments showed that choice of right measure allowed personalized reasoning models to provide better results for just 4 annotations per user compared to Random selection with 14 annotations per user.

The best results on the *WikiDetox Toxicity* data set (the middle in Fig. 5) were observed for the Controversy measure. However, the Var Ratio also performed better than the Random assignment. The Conformity Distance and Average Minimal Conformity turned out not to be the most stable but still slightly better for random baseline, especially in the range of 3-6/8 annotations per user. Stranger Count measure performed the worst.

For the *Unhealthy Conversations* dataset (the bottom plot in Fig. 5), the best results were achieved by the Controversy measure. However, the Average Minimal Conformity and the Ratio Distance still performed better than the Random selection. Surprisingly, the Var Ratio measure, which performed very well on the *WikiDetox* datasets, provided worse results than the Random baseline. It may be related to the high variance between the labels, which affects the final measure values.

Choosing the best PALS measure for data annotation (Controversy) allowed our model to outperform even by more than 30%, Fig. 6. It means that better selected training cases significantly boost quality of personalized reasoning since they deliver more information about user diversity. Moreover, using the Controversy or the Var Ratio measure allowed for achieving the similar results as for the Random assignment, but using only about 25% of the data, Fig. 7. In other words, PALS methods boost training so that it requires much less learning samples and computing resources.

The detailed results on all three datasets and the F1 score values for all eight labels in the *Unhealthy Conversations* dataset are presented in Appendix A.

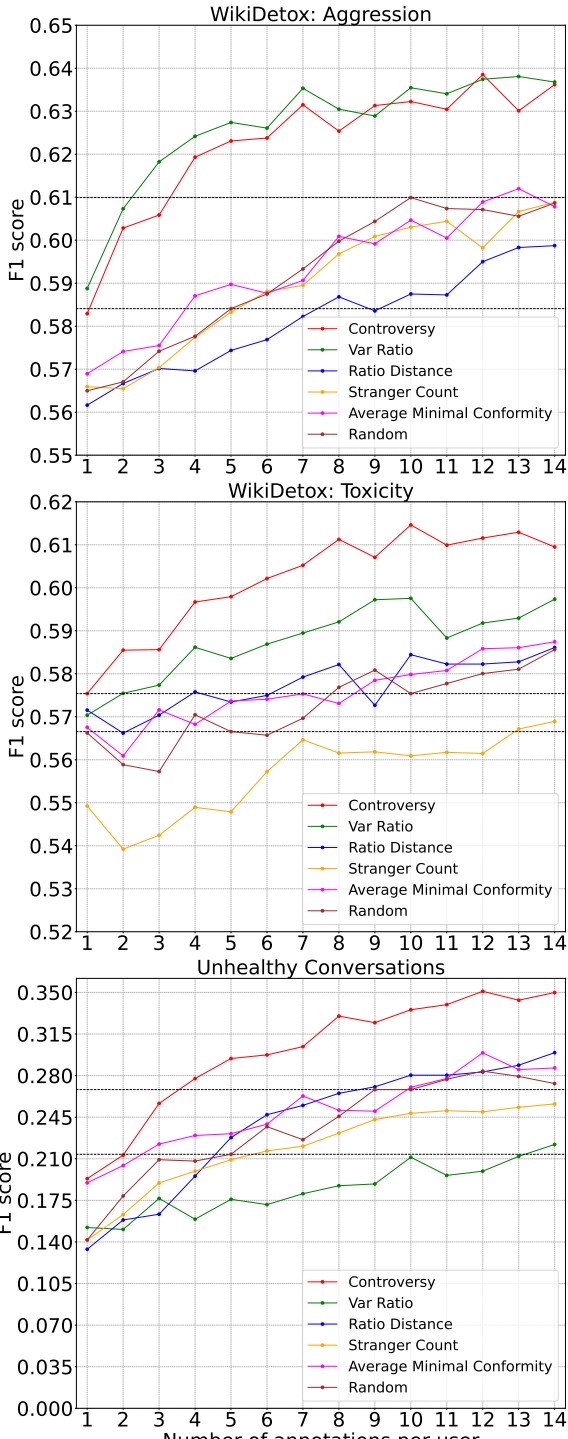

Figure 5: Performance of the models trained on data acquired using various active learning methods of text assignment. Y axis corresponds to F1 score for the aggressive or toxicity class for *WikiDetox* data and to F1 for positive class averaged over all classes for *Unhealthy Conversations*. Two horizontal lines indicate the reference performance (random selection) for 5 and 10 documents annotated by each user.

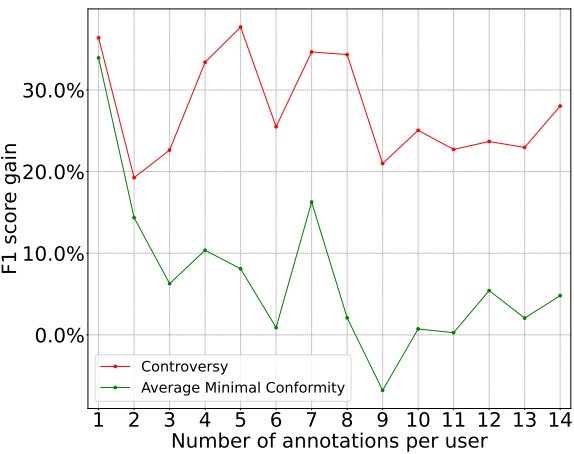

Figure 6: Gains in F1 score for *Unhealthy Conversations* and for our two active learning measures compared to Random selection. The Y axis denotes the gain of F1 for the positive class averaged over all classes.

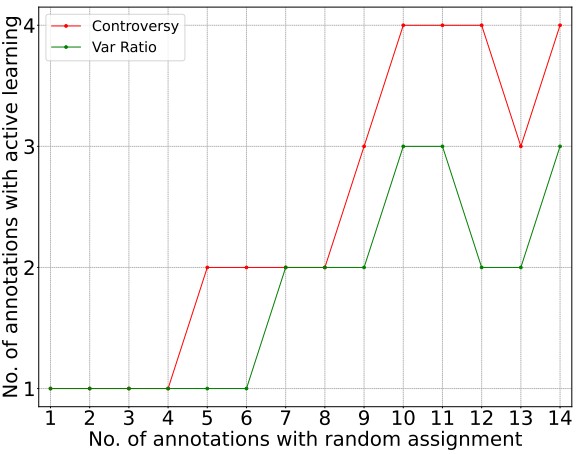

Figure 7: The least number of annotations needed for a given active learning measure (Y axis) to achieve a F1 score for positive class not worse than for the Random selection (X axis); the *WikiDetox: Aggression* dataset.

## 8 Discussion

By analyzing the results obtained on three datasets, we can come to the conclusion that the effectiveness of the model grows quickly with the addition of subsequent annotations for the users only up to a certain level, see Fig. 5. After it is exceeded, adding more samples to the set improves the performance only slightly, until it finally ceases to affect its performance at all. In the case of the *WikiDetox Aggression* dataset, the greatest increase in the efficiency of the model occurs in the range from 1 to 5 annotations per user, then the increase slows down until about 10 annotations is very insignificant. For *WikiDetox Toxicity*, the performance increases for up to 8 annotations per user, and then it fluctuates

around a certain value. A similar effect is observable for *Unhealthy Conversations* but the inflection point occurs later – at 12 texts. We can also see that the method of selecting texts for annotation also affects the amount of data needed to reach saturation point. Improving the quality of annotated texts still in the collection process reduces not only annotation costs but also training time, which positively impacts on the reduction of the carbon footprint emitted while training of deep learning models.

Most Active Learning approaches focus on the confidence of the model on unlabeled data. This works well for standard NLP methods where the text is the only source of knowledge. In this work, we show that, in personalized models, the subjectivity of people and their individual approach is also an essential factor that should be taken into account when selecting data samples for annotation. By selecting controversial texts and annotating them by different people, we gain more knowledge about their subjectivity, and therefore we improve the model quality of inference for these people.

A common method of evaluating NLP models is counting metrics globally on all annotations of individuals or on majority votes of annotators. However, in this way we lose information about specific individuals or groups of people for whom our model gives significantly different results. For this reason, we can consider evaluating models for individual annotators and then averaging these results or comparing their distributions.

## 9 Conclusions and Future Work

In this paper, for the first time, we raised a complex question: how to carry out the annotation process wisely and efficiently to reduce costs and/or to increase reasoning quality for subjective tasks solved in the personalized way?

Instead of commonly used random selection of texts to be annotated, we propose a novel paradigm: PALS – personalized active learning methods suitable for subjective NLP tasks. They allow us to efficiently gather information about user preferences related to text perception.

Experimental studies on three datasets and ten problems revealed that the Controversy measure appears to be the best out of all 20+ measures considered (only five best are described in the paper). The results show that PALS methods increase quality of collected data and are better suited both to the task and to the user. As a result, the classifiers

are able to improve their reasoning quality even by 30%. Alternatively, PALS methods can be used to reduce amount of data needed to reach a given performance; just 25%-40% is enough. Thus, models using data provided by PALS methods can learn in an extremely short time. This has a direct impact on reducing the costs of the entire annotation process.

In the future work, we will focus on the problem of text diversity, collection representativeness for the task, and reinforcement learning methods that would adjust the selection process runtime.

The code for all methods and experiments is publicly available [2] under the MIT license.

## Acknowledgments

This work was financed by (1) the National Science Centre, Poland, project no. 2021/41/B/ST6/04471; (2) Contribution to the European Research Infrastructure 'CLARIN ERIC - European Research Infrastructure Consortium: Common Language Resources and Technology Infrastructure', 2022-23 (CLARIN Q); (3) the Polish Ministry of Education and Science, CLARIN-PL; (4) the European Regional Development Fund, as a part of the 2014-2020 Smart Growth Operational Programme, projects no. POIR.04.02.00-00C002/19, POIR.01.01.01-00-0923/20, POIR.01.01.01-00-0615/21, and POIR.01.01.01-00-0288/22; (5) the statutory funds of the Department of Artificial Intelligence, Wroclaw University of Science and Technology; (6) the Polish Ministry of Education and Science within the programme "International Projects Co-Funded"; (7) the European Union under the Horizon Europe, grant no. 101086321 (OMINO). However, the views and opinions expressed are those of the author(s) only and do not necessarily reflect those of the European Union or the European Research Executive Agency. Neither the European Union nor European Research Executive Agency can be held responsible for them.

## 10 Limitations

The problem with testing Personalized Active Learning methods for subjective NLP tasks is the small number of relevant datasets. In particular, it is important that the dataset has a large number of annotations per text, and additionally a large number of annotations made by each user. Such properties would allow us to simulate various strategies of personalized selection texts for annotation. This would also facilitate studies on application of reinforcement learning methods to text selection, e.g., recalculation of PALS measures after each bulk data. To some extent, our results may depend on the way the data was collected. For example, in Wiki data, less offensive texts (with smaller number of offensive annotations) received less annotations in total than offensive ones. As a result, we plan to collect yet another data set with greater (possible full) coverage of annotations for each text. Moreover, due to limited computational resources, we did not consider fine-tuning the language model, and we were not able to try other language models. In addition, in our scenario, we assume that we have a certain data sample from the beginning to calculate the evaluated measures. The scenario considered is based on adding new annotations for each user proposed by our method. This is, to some extent, a suitable setup for incremental learning. However, we have not investigated any solutions dedicated to incremental learning - all our models were re-trained on the full, increased dataset. Furthermore, we tested many more measures (more than 20), but we have only included those in the paper that provided relatively better results. Moreover, we have not considered the diversity of the user community. Perhaps the datasets are biased towards some social groups. We also did not consider the context of data collection, e.g. the user mood, the broader context of the statement, the sequence of annotations in the case of analyzing the impact of the annotation order on the user perception, etc. Both limitations are due to the nature of the available data - the datasets did not included appropriate information. Moreover, we did not conduct a broader analysis of our models vulnerability to attacks. On the other hand, the quality and reliability of the data, e.g. potential mistakes in the annotation process, has not been analyzed and may have an impact on the results.

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

## A  Appendix

The results for the *WikiDetox Aggression* and *WikiDetox Toxicity* datasets are shown in Tab. 2 and Tab. 3 respectively. The results for each separate class in *Unhealthy Conversations* dataset are presented in Figures 8-14. The evaluation results averaged over all classes for the *Unhealthy Conversations* dataset are presented in Tab. 4. The class-specific values are described in Tables 5-11. The gains provided by our measures in comparison to the Random assignment for *WikiDetox Aggression* and *WikiDetox Toxicity* datasets are presented in Fig. 15 and Fig. 16 respectively. The least number of annotations needed for our measures to obtain equal or better results than the Random assignment for *WikiDetox Toxicity* and *Unhealthy Conversations* datasets are shown in Fig. 17 and Fig. 18 respectively.

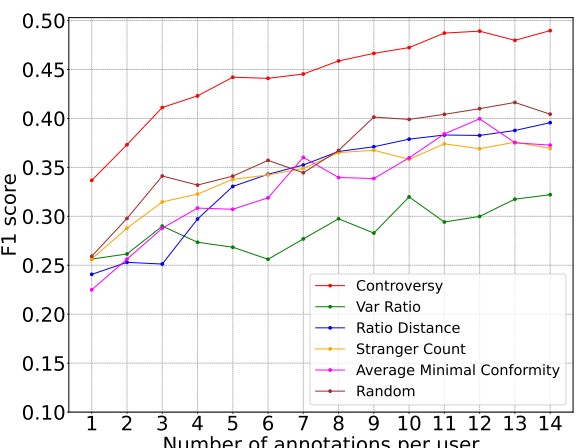

Figure 8: Results for *Unhealthy Conversations* and class *antagonize*.

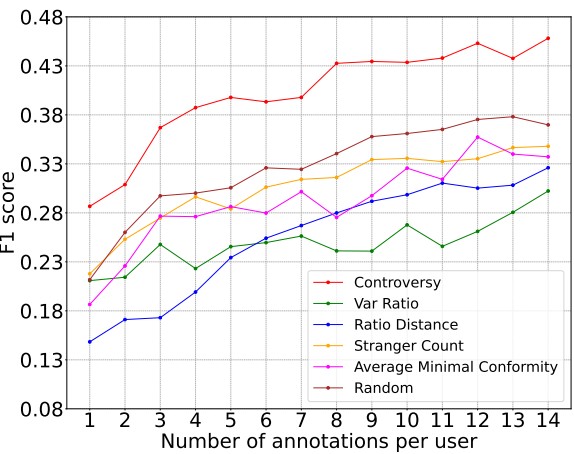

Figure 9: Results for *Unhealthy Conversations* and class *condescending*.

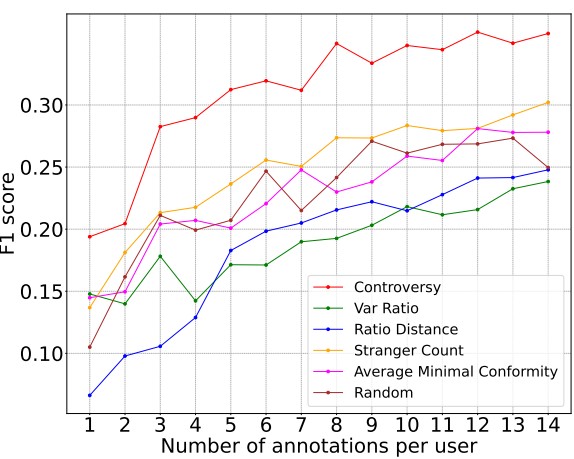

Figure 10: Results for *Unhealthy Conversations* and class *dismissive*.

| Measure | Number of annotations per user | | | | | | | | | | | | | |
|---|---|---|---|---|---|---|---|---|---|---|---|---|---|---|
| | 1 | 2 | 3 | 4 | 5 | 6 | 7 | 8 | 9 | 10 | 11 | 12 | 13 | 14 |
| Controversy | 58.29 | 60.28 | 60.59 | 61.93 | 62.31 | 62.38 | 63.15 | 62.54 | **63.13** | 63.23 | 63.04 | **63.85** | 63.01 | 63.62 |
| Var Ratio | **58.88** | **60.73** | **61.82** | **62.42** | **62.74** | **62.61** | **63.53** | **63.05** | 62.89 | **63.55** | **63.41** | 63.74 | **63.81** | 63.68 |
| Ratio Distance | 56.16 | 56.67 | 57.02 | 56.96 | 57.44 | 57.69 | 58.23 | 58.68 | 58.36 | 58.75 | 58.73 | 59.50 | 59.83 | 59.87 |
| Stranger Count | 56.59 | 56.55 | 57.04 | 57.74 | 58.33 | 58.81 | 58.95 | 59.68 | 60.09 | 60.30 | 60.44 | 59.82 | 60.67 | 60.88 |
| Average Minimal Conformity | 56.89 | 57.41 | 57.55 | 58.71 | 58.97 | 58.77 | 59.06 | 60.09 | 59.92 | 60.47 | 60.05 | 60.89 | 61.20 | 60.78 |
| Random | 56.50 | 56.71 | 57.42 | 57.77 | 58.41 | 58.75 | 59.33 | 59.98 | 60.44 | 60.99 | 60.74 | 60.71 | 60.56 | 60.86 |

Table 2: Results on *WikiDetox Aggression* dataset, for different number of annotations per user. **Bold** indicates the best result for a given number of annotations. Underlining indicates the best result for a given measure.

| Measure | Number of annotations per user | | | | | | | | | | | | | |
|---|---|---|---|---|---|---|---|---|---|---|---|---|---|---|
| | 1 | 2 | 3 | 4 | 5 | 6 | 7 | 8 | 9 | 10 | 11 | 12 | 13 | 14 |
| Controversy | **57.54** | **58.55** | **58.56** | **59.67** | **59.79** | **60.22** | **60.22** | **60.22** | **60.71** | 61.46 | **60.99** | **61.16** | **61.29** | **60.95** |
| Var Ratio | 57.04 | 57.55 | 57.74 | 58.62 | 58.36 | 58.69 | 58.95 | 59.21 | 59.72 | 59.76 | 58.83 | 59.18 | 59.30 | 59.74 |
| Ratio Distance | 57.15 | 56.62 | 57.04 | 57.58 | 57.34 | 57.50 | 57.92 | 58.22 | 57.27 | 58.44 | 58.23 | 58.23 | 58.28 | 58.61 |
| Stranger Count | 54.92 | 53.92 | 54.24 | 54.89 | 54.79 | 55.72 | 56.46 | 56.16 | 56.18 | 56.09 | 56.17 | 56.15 | 56.72 | 56.89 |
| Average Minimal Conformity | 56.75 | 56.09 | 57.16 | 56.82 | 57.37 | 57.41 | 57.53 | 57.31 | 57.85 | 57.98 | 58.08 | 58.58 | 58.61 | 58.74 |
| Random | 56.62 | 55.89 | 55.73 | 57.05 | 56.65 | 56.57 | 56.97 | 57.69 | 58.08 | 57.54 | 57.77 | 58.00 | 58.11 | 58.56 |

Table 3: Results on *WikiDetox Toxicity* dataset, for different number of annotations per user. **Bold** indicates the best result for a given number of annotations. Underlining indicates the best result for a given measure.

| Measure | Number of annotations per user | | | | | | | | | | | | | |
|---|---|---|---|---|---|---|---|---|---|---|---|---|---|---|
| | 1 | 2 | 3 | 4 | 5 | 6 | 7 | 8 | 9 | 10 | 11 | 12 | 13 | 14 |
| Controversy | **19.75** | **21.77** | **25.66** | **27.75** | **29.44** | **29.74** | **30.44** | **33.01** | **32.45** | **33.54** | **33.97** | 35.10 | **34.34** | **34.99** |
| Var Ratio | 15.22 | 15.05 | 17.66 | 15.91 | 17.59 | 17.14 | 18.06 | 18.73 | 18.88 | 21.13 | 19.60 | 19.96 | 21.20 | 22.20 |
| Ratio Distance | 13.65 | 16.20 | 16.82 | 19.83 | 23.10 | 24.89 | 25.90 | 26.72 | 26.95 | 28.22 | 27.90 | 28.35 | 28.87 | 29.89 |
| Stranger Count | 14.17 | 16.30 | 18.97 | 19.96 | 20.92 | 21.66 | 22.05 | 23.17 | 24.30 | 24.83 | 25.04 | 24.95 | 25.33 | 25.62 |
| Average Minimal Conformity | 18.98 | 20.43 | 22.23 | 22.96 | 23.11 | 23.91 | 26.28 | 25.08 | 25.00 | 27.02 | 27.76 | 29.91 | 28.50 | 28.64 |
| Random | 14.17 | 17.87 | 20.92 | 20.80 | 21.38 | 23.70 | 22.60 | 24.57 | 26.82 | 26.82 | 27.68 | 28.38 | 27.93 | 27.33 |

Table 4: Results F1 averaged for all classes on *Unhealthy Conversations* dataset, for different number of annotations per user. **Bold** indicates the best result for a given number of annotations. Underlining indicates the best result for a given measure.

| Measure | Number of annotations per user | | | | | | | | | | | | | |
|---|---|---|---|---|---|---|---|---|---|---|---|---|---|---|
| | 1 | 2 | 3 | 4 | 5 | 6 | 7 | 8 | 9 | 10 | 11 | 12 | 13 | 14 |
| Controversy | **33.67** | **37.32** | **41.12** | **42.30** | **44.21** | **44.10** | **44.54** | **45.87** | **46.64** | **47.24** | **48.72** | **48.91** | **47.98** | 48.97 |
| Var Ratio | 25.63 | 26.15 | 29.00 | 27.35 | 26.85 | 25.61 | 27.69 | 29.75 | 28.29 | 31.98 | 29.42 | 29.99 | 31.75 | 32.21 |
| Ratio Distance | 24.07 | 25.30 | 25.13 | 29.72 | 33.05 | 34.30 | 35.24 | 36.62 | 37.11 | 37.88 | 38.31 | 38.26 | 38.77 | 39.57 |
| Stranger Count | 25.65 | 28.79 | 31.47 | 32.27 | 33.78 | 34.22 | 34.89 | 36.51 | 36.75 | 35.82 | 37.41 | 36.90 | 37.57 | 36.94 |
| Average Minimal Conformity | 22.49 | 25.63 | 28.79 | 30.84 | 30.72 | 31.89 | 36.02 | 33.97 | 33.86 | 35.97 | 38.41 | 39.96 | 37.52 | 37.27 |
| Random | 25.93 | 29.78 | 34.12 | 33.19 | 34.10 | 35.72 | 34.45 | 36.73 | 40.14 | 39.90 | 40.42 | 40.99 | 41.63 | 40.43 |

Table 5: Results of F1 for class *antagonize* on *Unhealthy Conversations* dataset, for different number of annotations per user. **Bold** indicates the best result for a given number of annotations. Underlining indicates the best result for a given measure.

| Measure | Number of annotations per user | | | | | | | | | | | | | |
|---|---|---|---|---|---|---|---|---|---|---|---|---|---|---|
| | 1 | 2 | 3 | 4 | 5 | 6 | 7 | 8 | 9 | 10 | 11 | 12 | 13 | 14 |
| Controversy | **28.67** | **30.88** | **36.69** | **38.73** | **39.78** | **39.33** | **39.78** | **43.26** | **43.45** | **43.36** | **43.80** | **45.31** | **43.76** | 45.81 |
| Var Ratio | 21.08 | 21.43 | 24.77 | 22.31 | 24.55 | 24.96 | 25.63 | 24.12 | 24.09 | 26.77 | 24.58 | 26.11 | 28.05 | 30.23 |
| Ratio Distance | 14.83 | 17.11 | 17.30 | 19.92 | 23.43 | 25.41 | 26.70 | 27.99 | 29.18 | 29.85 | 31.04 | 30.52 | 30.83 | 32.60 |
| Stranger Count | 21.78 | 25.30 | 27.47 | 29.64 | 28.40 | 30.62 | 31.42 | 31.61 | 33.43 | 33.56 | 33.23 | 33.52 | 34.66 | 34.79 |
| Average Minimal Conformity | 18.65 | 22.58 | 27.67 | 27.61 | 28.63 | 27.98 | 30.16 | 27.54 | 29.75 | 32.56 | 31.42 | 35.73 | 34.00 | 33.72 |
| Random | 21.18 | 26.01 | 29.73 | 30.02 | 30.56 | 32.59 | 32.44 | 34.05 | 35.78 | 36.10 | 36.52 | 37.53 | 37.81 | 36.98 |

Table 6: Results of F1 for class *condescending* on *Unhealthy Conversations* dataset, for different number of annotations per user. **Bold** indicates the best result for a given number of annotations. Underlining indicates the best result for a given measure.

| Measure | Number of annotations per user | | | | | | | | | | | | | |
|---|---|---|---|---|---|---|---|---|---|---|---|---|---|---|
| | 1 | 2 | 3 | 4 | 5 | 6 | 7 | 8 | 9 | 10 | 11 | 12 | 13 | 14 |
| Controversy | 12.09 | 15.42 | 15.77 | **18.53** | **20.83** | 21.75 | 22.77 | 24.43 | 23.73 | 24.36 | 25.21 | 25.75 | **26.63** | 25.61 |
| Var Ratio | 4.75 | 5.42 | 7.75 | 6.82 | 10.16 | 8.64 | 9.55 | 9.20 | 12.06 | 13.41 | 12.23 | 12.54 | 12.07 | 13.55 |
| Ratio Distance | 14.23 | 15.14 | 15.31 | 18.25 | 19.82 | **22.59** | **24.35** | **26.06** | **25.15** | **26.23** | **26.21** | **26.49** | 26.59 | 28.45 |
| Stranger Count | 5.59 | 7.08 | 10.79 | 11.12 | 11.18 | 11.30 | 12.45 | 12.50 | 13.89 | 16.70 | 15.27 | 15.98 | 15.73 | 16.15 |
| Average Minimal Conformity | **21.62** | **21.27** | **16.65** | 17.84 | 20.28 | 19.43 | 21.35 | 20.92 | 21.45 | 22.67 | 22.72 | 23.78 | 23.33 | 23.53 |
| Random | 5.00 | 6.73 | 9.50 | 9.79 | 11.41 | 12.47 | 10.69 | 13.48 | 16.27 | 15.72 | 15.87 | 17.99 | 16.50 | 16.55 |

Table 7: Results of F1 for class *generalisation* on *Unhealthy Conversations* dataset, for different number of annotations per user. **Bold** indicates the best result for a given number of annotations. Underlining indicates the best result for a given measure.

| Measure | Number of annotations per user | | | | | | | | | | | | | |
|---|---|---|---|---|---|---|---|---|---|---|---|---|---|---|
| | 1 | 2 | 3 | 4 | 5 | 6 | 7 | 8 | 9 | 10 | 11 | 12 | 13 | 14 |
| Controversy | 8.15 | 9.94 | 11.96 | 14.39 | 14.27 | 17.29 | 18.81 | 22.06 | 20.57 | 22.43 | 21.98 | 22.78 | 24.11 | 23.05 |
| Var Ratio | 3.83 | 3.43 | 4.29 | 3.85 | 5.22 | 5.48 | 5.90 | 6.35 | 7.33 | 8.78 | 9.43 | 8.26 | 9.15 | 10.46 |
| Ratio Distance | 14.08 | 16.82 | 16.07 | **19.36** | **19.61** | **22.68** | **23.12** | **23.24** | **23.47** | **25.65** | **24.38** | **25.17** | 25.61 | 26.09 |
| Stranger Count | 4.34 | 4.55 | 6.65 | 8.82 | 9.42 | 9.11 | 9.22 | 10.13 | 10.95 | 11.87 | 11.32 | 10.94 | 11.79 | 12.20 |
| Average Minimal Conformity | **23.21** | **20.92** | **17.52** | 19.10 | 16.23 | 18.53 | 17.57 | 18.03 | 15.72 | 17.93 | 19.09 | 20.59 | 18.08 | 20.29 |
| Random | 3.59 | 5.22 | 6.90 | 7.04 | 7.62 | 10.05 | 9.21 | 11.35 | 14.73 | 12.47 | 14.00 | 14.98 | 13.29 | 13.20 |

Table 8: Results of F1 for class *generalisation unfair* on *Unhealthy Conversations* dataset, for different number of annotations per user. **Bold** indicates the best result for a given number of annotations. Underlining indicates the best result for a given measure.

| Measure | Number of annotations per user | | | | | | | | | | | | | |
|---|---|---|---|---|---|---|---|---|---|---|---|---|---|---|
| | 1 | 2 | 3 | 4 | 5 | 6 | 7 | 8 | 9 | 10 | 11 | 12 | 13 | 14 |
| Controversy | 9.76 | 10.04 | 16.10 | 19.18 | 21.13 | 19.58 | 20.68 | 23.67 | 23.37 | 24.02 | 25.09 | 27.15 | 24.71 | 26.90 |
| Var Ratio | 13.35 | 14.98 | 17.67 | 16.82 | 19.87 | 18.32 | 18.07 | 20.75 | 19.39 | 21.44 | 18.64 | 19.73 | 20.79 | 22.02 |
| Ratio Distance | 7.09 | 11.07 | 14.76 | 18.65 | 23.30 | **24.13** | 25.12 | 25.16 | **25.62** | 29.04 | 25.82 | 26.50 | 28.68 | 27.97 |
| Stranger Count | 12.27 | 13.03 | 15.74 | 16.89 | 19.96 | 20.39 | 21.07 | 21.44 | 24.20 | 23.82 | 25.69 | 24.71 | 24.02 | 24.78 |
| Average Minimal Conformity | **18.90** | **20.70** | **24.01** | **23.05** | **23.64** | 24.07 | **25.90** | **25.44** | 23.60 | 27.17 | 27.82 | **28.69** | **28.95** | **28.00** |
| Random | 12.99 | 16.61 | 18.52 | 19.84 | 19.28 | 22.34 | 22.42 | 24.18 | 23.03 | 25.34 | **27.84** | 27.88 | 25.86 | 26.96 |

Table 9: Results of F1 for class *sarcastic* on *Unhealthy Conversations* dataset, for different number of annotations per user. **Bold** indicates the best result for a given number of annotations. Underlining indicates the best result for a given measure.

| Measure | Number of annotations per user | | | | | | | | | | | | | |
|---|---|---|---|---|---|---|---|---|---|---|---|---|---|---|
| | 1 | 2 | 3 | 4 | 5 | 6 | 7 | 8 | 9 | 10 | 11 | 12 | 13 | 14 |
| Controversy | **23.54** | **25.10** | **29.71** | **32.14** | **34.62** | **34.21** | **35.30** | **36.80** | **36.06** | **38.59** | **38.55** | **39.90** | **38.24** | **38.82** |
| Var Ratio | 23.09 | 19.98 | 22.31 | 20.02 | 19.34 | 19.87 | 20.59 | 21.68 | 20.69 | 23.68 | 21.74 | 21.50 | 23.36 | 23.12 |
| Ratio Distance | 12.76 | 15.73 | 15.20 | 18.00 | 22.00 | 23.99 | 23.41 | 24.95 | 26.65 | 26.11 | 27.70 | 27.10 | 27.50 | 30.05 |
| Stranger Count | 15.85 | 17.23 | 19.32 | 19.20 | 20.04 | 20.43 | 20.24 | 22.61 | 23.52 | 23.67 | 24.47 | 24.47 | 24.36 | 24.25 |
| Average Minimal Conformity | 13.48 | 16.96 | 20.61 | 21.57 | 22.17 | 23.41 | 28.19 | 26.70 | 26.82 | 26.91 | 29.32 | 32.55 | 29.86 | 29.88 |
| Random | 20.00 | 24.56 | 26.57 | 25.84 | 25.95 | 28.06 | 27.52 | 28.05 | 30.73 | 32.09 | 32.31 | 32.39 | 33.10 | 32.22 |

Table 10: Results of F1 for class *hostile* on *Unhealthy Conversations* dataset, for different number of annotations per user. **Bold** indicates the best result for a given number of annotations. Underlining indicates the best result for a given measure.

| Measure | Number of annotations per user | | | | | | | | | | | | | |
|---|---|---|---|---|---|---|---|---|---|---|---|---|---|---|
| | 1 | 2 | 3 | 4 | 5 | 6 | 7 | 8 | 9 | 10 | 11 | 12 | 13 | 14 |
| Controversy | **19.39** | **20.45** | **28.26** | **28.98** | **31.24** | **31.94** | **31.19** | **34.95** | **33.36** | **34.79** | **34.45** | 35.87 | **34.97** | **35.75** |
| Var Ratio | 14.78 | 13.99 | 17.82 | 14.23 | 17.14 | 17.12 | 18.99 | 19.26 | 20.32 | 21.82 | 21.16 | 21.58 | 23.25 | 23.84 |
| Ratio Distance | 6.62 | 9.79 | 10.57 | 12.89 | 18.28 | 19.84 | 20.50 | 21.55 | 22.21 | 21.48 | 22.78 | 24.12 | 24.16 | 24.78 |
| Stranger Count | 13.68 | 18.12 | 21.33 | 21.76 | 23.64 | 25.57 | 25.05 | 27.36 | 27.34 | 28.35 | 27.93 | 28.11 | 29.20 | 30.21 |
| Average Minimal Conformity | 14.48 | 14.96 | 20.41 | 20.71 | 20.08 | 22.06 | 24.77 | 22.99 | 23.81 | 25.89 | 25.53 | 28.10 | 27.79 | 27.81 |
| Random | 10.51 | 16.16 | 21.11 | 19.92 | 20.72 | 24.67 | 21.50 | 24.16 | 27.08 | 26.12 | 26.84 | 26.87 | 27.33 | 24.96 |

Table 11: Results of F1 for class *dismissive* on *Unhealthy Conversations* dataset, for different number of annotations per user. **Bold** indicates the best result for a given number of annotations. Underlining indicates the best result for a given measure.

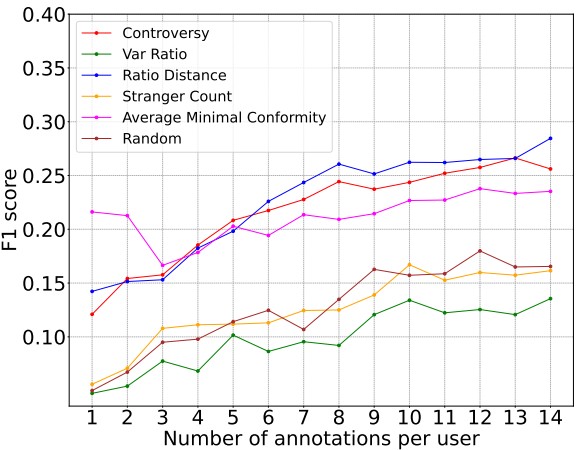

Figure 11: Results for *Unhealthy Conversations* and class *generalisation*.

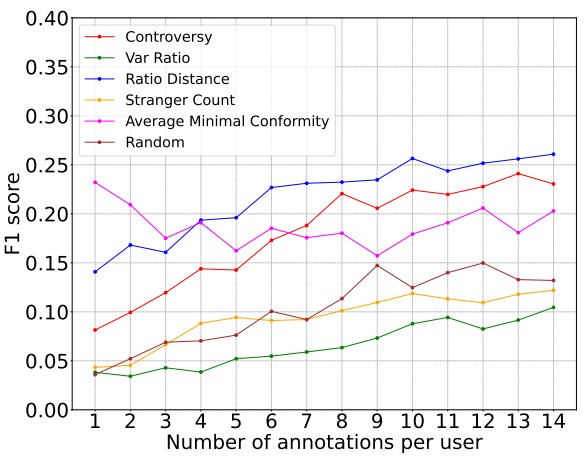

Figure 12: Results for *Unhealthy Conversations* and class *generalisation unfair*.

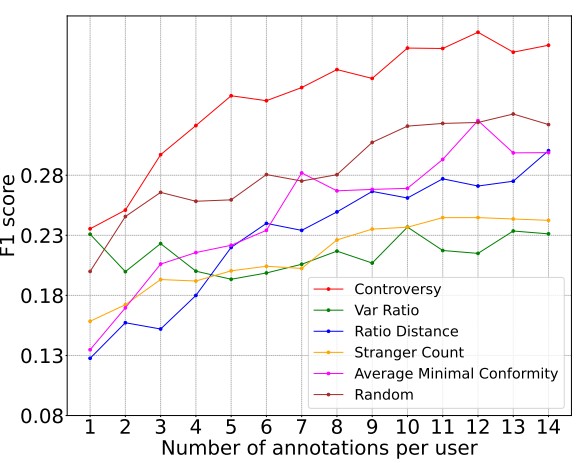

Figure 13: Results for *Unhealthy Conversations* and class *hostile*.

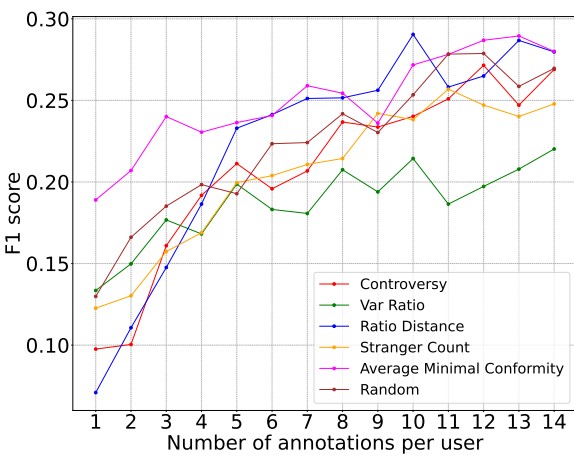

Figure 14: Results for *Unhealthy Conversations* and class *sarcastic*.

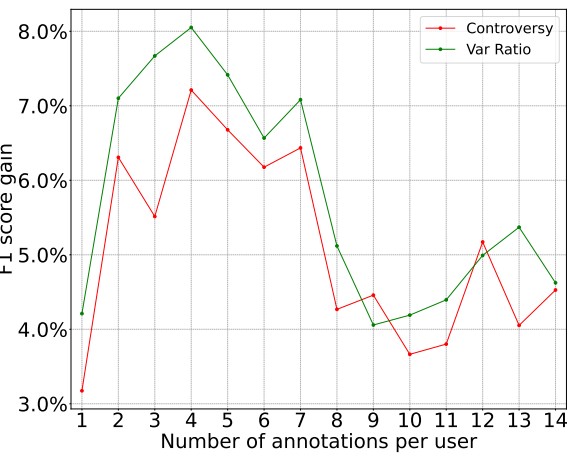

Figure 15: Gains for *WikiDetox: Aggression*. The Y axis shows the gain of F1 score for the positive class with respect to Random selection.

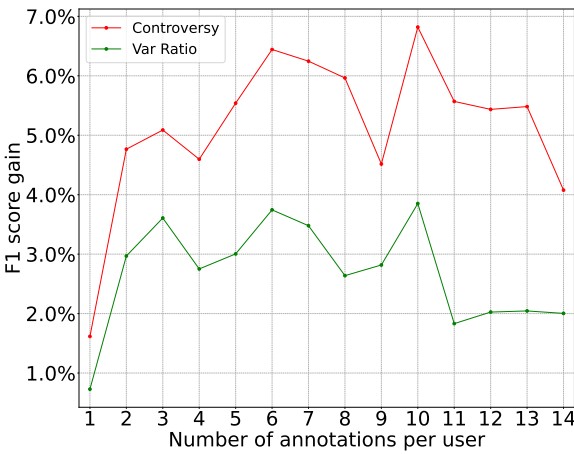

Figure 16: Gains for *WikiDetox: Toxicity*. The Y axis shows the gain of F1 score for the positive class with respect to Random assignment.

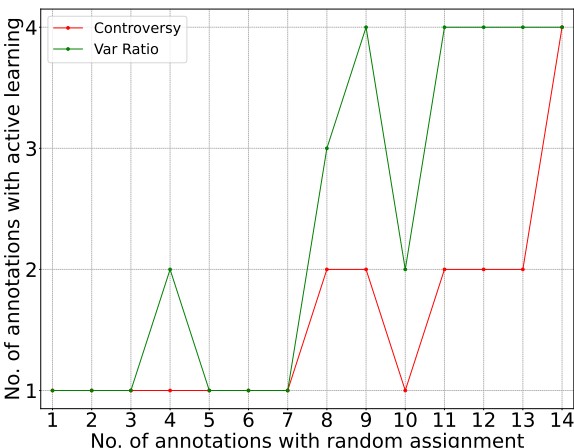

Figure 17: The least number of annotations needed for a given measure (Y axis) to achieve a F1 score for the positive class not less than the Random selection (X axis) for the *WikiDetox: Toxicity* dataset.

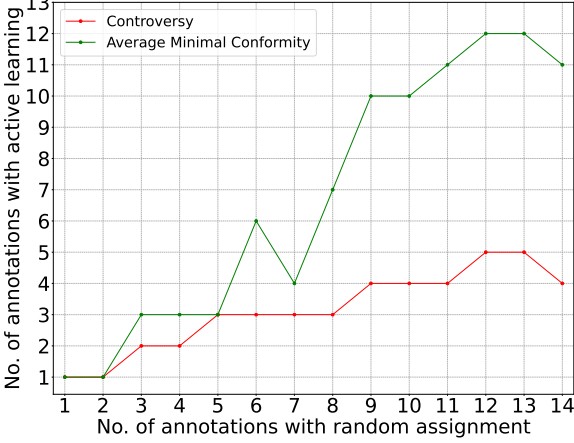

Figure 18: The least number of annotations needed for a given measure (Y axis) to achieve a F1 score for the positive class averaged across all classes not less than the Random selection (X axis) for the *Unhealthy Conversations* dataset.