# OpenReview forum: "PALS: Personalized Active Learning for Subjective Tasks in NLP"
_EMNLP/2023/Conference — EMNLP 2023 Main_

### Official Review · Reviewer_FazH · 2023-07-25

**Soundness:** 4

**Excitement:**

4: Strong: This paper deepens the understanding of some phenomenon or lowers the barriers to an existing research direction.

**Paper Topic And Main Contributions:**

The authors present a method to select data for annotation for subjective tasks. Because the task is subjective, the threshold for what each person considers "acceptable" differs, so the proposed method chooses data using active learning so that the threshold for each person is more accurately determined given fewer annotations. The method is validated on three common datasets for subjective tasks. The results show that the selected data annotations yield better accuracy, even with fewer annotations.

**Reasons To Accept:**

The results show a way to reduce the number of annotations required for subjective tasks (many of which can be psychologically damaging) while maintaining or even increasing accuracy. It is also important to bring attention to the problems with labeling for subjective tasks.

**Reasons To Reject:**

As mentioned in the Limitations section, testing the method on a broader variety of datasets with more diverse annotators remains as future work.

**Reproducibility:**

4: Could mostly reproduce the results, but there may be some variation because of sample variance or minor variations in their interpretation of the protocol or method.

**Reviewer Confidence:**

4: Quite sure. I tried to check the important points carefully. It's unlikely, though conceivable, that I missed something that should affect my ratings.

**Typos Grammar Style And Presentation Improvements:**

I recommend going through the text with a copy editor as there are several grammar errors. For example, in the abstract, line 19 is missing an apostrophe, line 23 is missing the word "the", and line 24 has an extra "even".

---

> ### Author Rebuttal · Authors · 2023-08-29
>
> Dear Reviewer,
>
> Thank you for your valuable suggestions and pointing out the risk of causing psychological damage to the users during the annotation process. This aspect is especially important in the case of annotation regarding the offensiveness of the texts as the abusive content may negatively affect the user's mental well-being.
>
> In the future work, we plan to evaluate our methods on diverse datasets with various groups of annotators.
>
> Moreover, we want to employ an experienced copy editor in order to revise the language, and correct the typos in the entire content to make it more understandable.

---

### Official Review · Reviewer_66GU · 2023-08-02

**Soundness:** 4

**Excitement:**

4: Strong: This paper deepens the understanding of some phenomenon or lowers the barriers to an existing research direction.

**Paper Topic And Main Contributions:**


In this paper, the authors explore the possibility of applying active learning technics to subjective tasks. The authors try to address an important question that how to choose text to annotate so as to not only reduce the total cost, but also improve the downstream task performances. To do this, the authors identify 5 measures, including controversy and ratio distance, as metrics to select data to be annotated. Then the authors experiment with the Wikipedia Detox project and Unhealthy Conversations dataset. Results show that PALS methods could outperform a random baseline and require fewer training samples.

In general, I think it is an interesting work. There is a growing trend that brings methods from the active learning community to NLP these days, such as choosing which data to be included in the prompt for large language models. I would rather treat this paper as very timely, although this paper is studying a relevant problem: personalized selection of annotating texts for subjective tasks. Regarding the designed measures, I think they fit into the general goal of incorporating the users' preferences. Experimental design follows the literature in active learning and experimental results could support their claims.

**Reasons To Accept:**


I think it is a reasonable step of applying active learning for NLP tasks. The motivation of this paper is clear. The authors provide a clear definition of the problem formulation, along with clear explanation for their proposed measures. The experimental results look fine.

**Reasons To Reject:**


I do not have major reasons to reject this paper. The authors have pointed out there are no prior works for this problem, so that is probably the reason why a random baseline is used. But since the goal of this paper is not to achieve the state of the art performances in a task, so it is totally fine and reasonable.

**Reproducibility:**

4: Could mostly reproduce the results, but there may be some variation because of sample variance or minor variations in their interpretation of the protocol or method.

**Reviewer Confidence:**

4: Quite sure. I tried to check the important points carefully. It's unlikely, though conceivable, that I missed something that should affect my ratings.

---

> ### Author Rebuttal · Authors · 2023-08-29
>
> Dear Reviewer,
>
> Thank you for your review and emphasizing the promising direction of our research. In our opinion, the annotation process for the subjective NLP tasks is often more expensive and prone to unnecessary expenses than the annotation process for the generalized NLP tasks. Therefore, the development of personalized methods for cost optimization could provide even higher cost reduction that the ones used in generalized NLP problems.
>
> In the camera-ready version of the paper we will also work on the language to make the article more understandable and readable.

---

### Official Review · Reviewer_X3Lx · 2023-08-04

**Soundness:** 3

**Excitement:**

4: Strong: This paper deepens the understanding of some phenomenon or lowers the barriers to an existing research direction.

**Paper Topic And Main Contributions:**

This paper presented a user-level active learning application. Unlike the generalized active learning approach, the authors claimed that their approach could improve the model's performance on subjective/personalized tasks. The authors compared five measures to select the training samples on two toxic conversation datasets. The results showed the 'controversy' (entropy-based measure) strategy works consistently better than other active learning and baseline strategies with their classifier.

**Questions For The Authors:**

A. Can you add justification for using the XLM-RoBERTa model than other pre-trained models?

B. Can you explain why particular strategies worked better/worse than others?

C. Do you have any results with the ensemble approach? e.g., using multiple measures to score each example.

**Reasons To Accept:**

- The paper was easy to read. The research questions were well justified, and the intuitions were effectively presented with figures.
- Adding a user-level dimension to the scoring measures for the active learning application is an interesting idea.

**Reasons To Reject:**

- Although the paper's main contribution is to suggest the benefit of personalized active learning methods, the analysis results lack comparison with generalized active learning strategies (e.g., ablation study without user dimensions).
- It is unclear how the authors split the data with a relative timeframe (i.e., past, present, future1, and future2).
- As I understand, the past set was used to build a background model, and the present set was used to update the model with different measures. However, Figure 4 shows that the past set includes the same user data from the future1 (dev) and the future2 (test) sets (in different timeframes). Is this a correct interpretation? If so, it would contradict the authors' intention of using user-level stratification (Section 5) for cross-validation design and cause some data leakage in the inference stage, as the model already knows something about users in the dev and test sets.
- What's the performance plateau for each dataset? (e.g., how far would the model reach when it is trained with the entire training dataset?)

**Reproducibility:**

3: Could reproduce the results with some difficulty. The settings of parameters are underspecified or subjectively determined; the training/evaluation data are not widely available.

**Reviewer Confidence:**

3: Pretty sure, but there's a chance I missed something. Although I have a good feel for this area in general, I did not carefully check the paper's details, e.g., the math, experimental design, or novelty.

**Typos Grammar Style And Presentation Improvements:**

- In Section 6.4 (lines 424-426), are the use of 'present' and 'pastes' swapped as a typo?
- I think the presentations of figures can be improved for better readability. For example, each strategy can be further distinguished with different line/point styles, or the baseline (random selection) can be shown in a dotted line, etc.

---

> ### Author Rebuttal · Authors · 2023-08-29
>
> Dear Reviewer,
>
> Thank you for your review. We appreciate your help and will do our best to apply your suggestions to improve the quality of the article.
>
> The main goal of our work was to design methods for optimizing the size of the train set. while keeping the performance similar to training on the full dataset. Personalization is a crucial approach to prediction of subjective perception of text. Hence, our second contribution was proposing the active learning methods to be used in subjective NLP tasks and our paper is the first to pose this problem. The main difference between general active learning methods used for non-personalized methods and personalized ones is respecting the user context instead of assuming the gold standard for each text. This is unique in the active learning domain, thus requires our dedicated methods.
>
> Having regard to the completeness we collected comparative results for non-personalized methods with similar setups, but it will not provide much insight related to contextualized data collection.
>
> Regarding the new texts scenario, our test set always contained texts that did not appear in the training set or the dev set. This allows us to simulate a situation when new texts appear in the system. According to the results of our experiments, the training samples provided by our methods allowed the models to train and achieve good results on the new texts in the test set. This is due to the fact that the model used the similarity of vector representations of texts, obtained from the language model. This topic is further explained in section 6.1 and in Figure 4.
> According to your interpretation, the Past set represents the general knowledge about all users. It is a starting knowledge source used to prepare a starting model for each of the proposed methods. In the next steps, each method was used to select one new text for each user from the Present set in every iteration of the Active Learning algorithm (see Algorithm 1). After every iteration, the model was trained on the enriched dataset and then evaluated on the same (constant) dev (Future 1) and test (Future 2) sets. The conducted evaluation leveraged the cross-validation mechanism, which guarantees the model will be tested on users, for which the only data source is the small Past set.
>
> The performance plateau for each dataset is similar to the one achieved by the best measure for the maximum number of annotations per user. That is why we selected the 14 users as the upper limit in presenting the results of our experiments. We will add appropriate information in the camera-ready version of the paper.
>
> Answer to Question A: We used English language models during experiments, but their impact on results was insignificant compared to the multilingual model. What is more, the XLM-RoBERTa model achieved the best results out of all multilingual models used during experiments. Moreover, the use of multilingual model shows the possibility of applying our methods to various other languages and the higher versatility of our approaches.
>
> Answer to Question B: We will add the more detailed analysis focused on explaining the results for each of the methods on the additional page in the camera-ready version of the paper. The higher values of the Var Ratio F1-score for the WikiDetox imply that the tasks in those datasets seem to be significantly easier to learn than the ones available in the Unhealthy Conversations dataset. In the case of the more difficult task, a simple intuition based on selecting the texts which users annotated with higher variability may prevent the model from gathering the basic  knowledge about the task and its relation to the language. These basic intuitions should be gathered from the texts, which are characterized by low variability of user annotations and they are considered as easy to learn and extract the most basic patterns useful for the task learned.
>
> Answer to Question C: As this is quite a novel topic and there are no prior works regarding this matter, we plan to leverage the ensemble architectures synthesizing knowledge obtained via various measures in the future work.
>
> Moreover, we will revise the language and correct the typos in the entire content to make it more understandable.

---

### Meta-Review · Area_Chair_Z4EY · 2023-09-15

**Recommendation:** 4

**Metareview:**

The reviewers generally agreed that the paper was well written and the experiments are sound. Reviewer X3Lx appreciated the user-level dimension to scoring. Reviewer FazH also pointed out that the results reduce the number of annotations required for subjective tasks, while keeping the accuracy level.

Reviewer X3Lx had most notable concerns including the lack of comparisons in findings.The authors mentioned conducting some analyses but suggested that they did it find useful insights. It was still a little unclear. Reviewer 66GU mentioned that they do not have a concern about a lack of a baseline. Reviewer FazH also thought that leaving testing of the method on a broader variety of datasets with more diverse annotators as a future work.

---

### Decision · Program_Chairs · 2023-10-07

**Decision:**

Accept-Main

**Comment:**

The reviewers generally agreed that the paper was well written and the experiments are sound. Reviewer X3Lx appreciated the user-level dimension to scoring. Reviewer FazH also pointed out that the results reduce the number of annotations required for subjective tasks, while keeping the accuracy level.

Reviewer X3Lx had most notable concerns including the lack of comparisons in findings.The authors mentioned conducting some analyses but suggested that they did it find useful insights. It was still a little unclear. Reviewer 66GU mentioned that they do not have a concern about a lack of a baseline. Reviewer FazH also thought that leaving testing of the method on a broader variety of datasets with more diverse annotators as a future work.